# Within-individual variation of HbA1c measurements in primary care: A retrospective cohort study

Alex Gough[1]*, Tom Marshall[1], Alice Sitch[1,2]

**1** Department of Applied Health Sciences, University of Birmingham, Edgbaston Campus, Birmingham, West Midlands, United kingdom, **2** National Institute for Health and Care Research (NIHR) Birmingham Biomedical Research Centre, Birmingham, United Kingdom

* a.gough.2@bham.ac.uk

## Abstract

### Background

HbA1c is a marker for diabetes mellitus that reflects average glucose concentrations over the previous eight to twelve weeks. It is used to aid in the diagnosis and management of diabetes. Variation in within-individual measured HbA1c may affect its clinical utility but estimates of this are based on limited data that is often not generalisable to real-world settings.

### Methods

A retrospective cohort study was performed using data on HbA1c results and socio-demographic, lifestyle and comorbidity covariates extracted from the IQVIA Medical Research Database (IMRD) database using the DEXTER tool. A minimum of four measurements in the same individual was the only inclusion criterion. Within-individual measured variation was calculated as a coefficient of variation (CV) using a linear regression random effects model for the whole population and various subgroups.

### Results

587,023 participants were included in this study, making it the largest study of variation of HbA1c to date. The overall measured within-individual coefficient of variation ($CV_T$) was 0.20 (95%CI 0.20 to 0.20). This is around three times higher than reported in a previous systematic review. $CV_T$ increased with patient mean HbA1c level.

### Strengths and limitations

The large number of participants and the real-world nature of the results are important strengths of this study. Weaknesses included the problem of accounting for confounding by indication.

**Data availability statement:** Data cannot be shared publicly because the authors do not have permission to share the data. IMRD data used for the study were obtained under licence from IQVIA; pseudonymised participant data are available from IQVIA subject to Scientific Review Committee approval. IMRD-UK data contains electronic health records from UK primary care. In compliance with the UK Data Protection Act and licensing agreements, the data cannot be shared via a public repository. These restrictions aim to protect patient confidentiality. Requests can be submitted to bassam.bafadhal@iqvia.com. The data underlying the results presented in the study are available from IQVIA (https://www.iqvia.com/locations/united-kingdom/solutions/life-sciences-industry-solutions/real-world-solutions/iqvia-medical-research-data).

**Funding:** The author(s) received no specific funding for this work.

**Competing interests:** No authors have competing interests.

## Conclusions

Estimated within-individual variation in this analysis of real-world data is very high and is higher than previously reported. Variation increases with patient mean HbA1c, that is with more severe disease status. This has important implications for the diagnosis, monitoring and clinical decision-making for diabetes.

## Introduction

Diabetes mellitus (DM) is a chronic metabolic disorder characterised by disturbances of carbohydrate, fat and protein metabolism [1]. Type 1 diabetes mellitus (T1DM) accounts for around 5–10% of cases, and tends to have a younger age of onset. Type 2 diabetes mellitus (T2DM) accounts for 90–95% of cases. It has a complex aetiology, and results in insulin resistance and relative insulin deficiency. Prediabetes refers to individuals who have glucose levels higher than normal but do not meet the criteria for diabetes [2].

Haemoglobin A1c (HbA1c) is formed from the non-enzymatic attachment of glucose to haemoglobin, and reflects average glucose levels over the preceding eight to twelve weeks [3]. It was previously commonly measured using National Glycohemoglobin Standardization Program (NGSP) reported in units of %, but more recent guidelines recommend it is measured using International Federation of Clinical Chemistry and Laboratory Medicine (IFCC) methods and reported in units of mmol/mol or in % derived from the established master equations [4].

HbA1c levels of 6.0% to 6.4% (42–47 mmol/mol) are consistent with a diagnosis of prediabetes, although the American Diabetes Association uses the lower value of 5.7% (39 mmol/mol) [5,6]. Levels of 6.5% (48 mmol/mol) or more are consistent with a diagnosis of diabetes, although age, ethnicity, gestational status and genetic factors can affect HbA1c levels [7].

However, HbA1c levels, like all biological measurements, vary within an individual over time. This within-individual variation is broadly composed of three parts. Pre-analytical variation is due to variation of factors prior to measurement, such as exercise, recent food or fluid intake or stress. Analytical variation is the variation introduced by imprecision in the measuring process. Biological variation is the within-individual variation, influenced partly by predictable factors such as season, time of day or monthly hormone cycles, but also to a large extent by chance. The sum of these three types of variation gives the total within-individual variation [8]. The higher the within-individual variation, the lower the probability that a single measurement is reflective of the true mean. In this study, it is not possible to separate analytical from biological variation, so within-individual coefficient of variation (CV) is taken to refer to total within-individual variation, that is the sum of biological and analytical variation.

The European Federation of Laboratory Medicine (EFLM) variation database lists the variation of common, laboratory-measured clinical tests [9]. However, most of its data comes from trials with small numbers of participants, often measured in ideal conditions. A recent systematic review [10] found 111 studies reporting variation of

HbA1c in a total of 564,143 participants. But most studies included few participants and/or small numbers of repeat measurements. This review also found that variation increases with mean HbA1c, and was higher in participants with T1DM and T2DM. However, the extent to which HbA1c variation differs between different subgroups such as age, health status, BMI and ethnicity is not known.

NICE guidelines suggest diabetes can be diagnosed with a single HbA1c measurement in patients with appropriate symptoms, although they suggest a repeat test is "sensible." In asymptomatic patients they recommend a second HbA1c measurement [11]. However, this advice only applies to initial diagnosis, not monitoring. Published studies on longitudinal testing of HbA1c tend to focus on an adequate number of tests being performed in a year, rather than repeat tests in the short-term to ascertain the accuracy of the test result, and in fact they sometimes actively discourage short-term repeat testing as wasteful of health resources [12,13].

This study aimed to use real-world data (that is data from a clinical source rather than an experimental setting) from the IQVIA Medical Research Database (IMRD) to estimate total within-individual variation of HbA1c in patients with and without diabetes and to compare this to previously published data. It explores the extent to which HbA1c variation differs with age, health status, BMI, ethnicity and other characteristics. It also considers the clinical implications of HbA1c variation in relation to diagnosis, monitoring and management of diabetes.

## Aims

To describe the within-individual variation of HbA1c in a large cohort of patients in a database of routine general practice records, and to describe any covariates that have an effect on the variation. To describe the probability of within-individual variation in HbA1c resulting in clinically important changes in HbA1c in order to inform clinical decision-making with regard to observed HbA1c changes in a patient.

## Methods

### Study design and data sources

A population-based retrospective cohort study was undertaken using data obtained from the IQVIA Medical Research Database (IMRD). IMRD incorporates data from The Health Information Network (THIN), a Cegedim Database. Reference made to THIN is intended to be descriptive of the data asset licensed by IQVIA. IMRD is a pseudanonymised database of the electronic healthcare records of patients registered with general practices in the UK which use compatible practice management systems. Data collection for the database commenced in 2003. As of March 2021, the database contained the record of over 20 million patients, of which around 4 million were active with their registered GP practice.

IMRD data has been shown to be generalisable to the general primary care population in terms of factors such as demographics, deprivation, condition prevalence and deaths [14]. The dataset used for this study consisted of 11,737,653 participants registered in 832 practices.

HbA1c tests in the database are the results of requests submitted from practices to central laboratories which have nationally mandated levels of quality control. The analysis method is not given, but most UK laboratories moved to the HPLC method after 2010. Analysis of the database shows that before 2010 most HbA1c levels were recorded as % and after 2011 most were recorded as mmol/mol, but some errors appear to have been made with units (see data cleaning).

This study is reported according to the STrengthening the Reporting of Observational studies in Epidemiology (STROBE) statement for observational studies.

### Ethical approval

IQVIA Medical Research data has been approved by the NHS Health Research Authority (NHS Research Ethics Committee ref 18/LO/0441) for the purpose of medical and public health research and to supply data for researchers for

scientifically approved studies. The use of the data for this study was approved by the IQVIA Scientific Review Committee on 19th November 2020 with the reference 20SRC068.

## Setting

Data was obtained from UK general practices registered with THIN/IMRD from database inception to 9th September 2021. Data extraction from the database was performed using the Dexter software [15] and was completed on 9th September 2021. The date of the latest entrance was 11th January 2021. The entire timeframe of the database from inception to last entry was used for analyses except for sensitivity analysis by year, which included only the years 2010–2019, since years prior to 2010 and after 2019 were incomplete and contained only small numbers of measurements. The authors had no access to data that could identify individual participants.

## Participants

All participants in the database were eligible for inclusion. Except for sensitivity analyses, participants were excluded if they did not have at least four repeat measurements of HbA1c expressed in mmol/mol recorded. Four was chosen as the cut-off for repeat measurements for pragmatic reasons related to maximising the number of repeat measurements in the analysis while allowing a sufficiently large sample size for analyses. Participants were followed until the earliest date of either death, leaving the practice, stopping contributing to the database or the study end date.

## Variable choice

The primary outcome was within-individual variation of HbA1c calculated as the total within-individual coefficient of variation ($CV_T$). Data on a number of other patient and test characteristics were also extracted: socio-demographic characteristics, diabetic status, diagnoses and comorbidities. Variables extracted that are included in the analyses are listed in Table 1. Variables were selected as those hypothesised most likely to have an effect on variation or that enabled calculation of variation (for example date of measurement). Code lists for the different variables are available at: https://github.com/THINKINGGroup/phenotypes.

## Sample size and bias

The study size included the entire database where participants had sufficient repeat measurements to reduce bias and to ensure that the sample sizes for the analyses were sufficiently large to help compensate for the weaknesses inherent in routinely collected medical data. The minimum number of participants for subgroup analysis was selected as 100, for pragmatic reasons, compromising between group size and number of subgroup analyses possible. Using simulation, (assuming a CV of 20%), 100 participants and 4 observations per participant provides an estimate with an approximate 95% confidence interval with a total width of less than 10% (+/- 5%).

## Data cleaning

When calculating coefficient of variation, mmol/mol and percentage are not interchangeable since one measure has an intercept on the y axis when converting to the other. Since most HbA1c measurements were reported in mmol/mol, those expressed as a percentage were excluded from the main analysis. Values of HbA1c greater than 195.1 mmol/mol were excluded as being implausible based on analysis of the IMRD data where fewer than 0.01% of measurements had values greater than 195.1 mmol/mol. Values of less than 20.1 mmol/mol were excluded as either implausible based on published data, or because they were possibly due to some unknown biological mechanism unrelated to diabetes [16]. However, a histogram of values <20.1 mmol/mol had a normal distribution around a median of 7.7 mmol/mol, values more consistent with units of per cent, suggesting most of these low values were actually per cent values erroneously recorded as mmol/mol. For BMI, implausible values were considered <14 or >70 [17] and these were re-categorised as missing.

**Table 1. Variables extracted from database.**

| Variables extracted | |
|---|---|
| Test characteristics | HbA1c result |
| | Date of measurement |
| | Unit of measurement |
| Patient characteristics | |
| Sociodemographic | Age |
| | Sex |
| | Ethnicity |
| | Townsend Deprivation Score |
| | BMI |
| | Geographical region |
| Lifestyle factors | Smoker status |
| | Alcohol consumption status |
| Diabetic status | Prediabetes |
| | Type one diabetes mellitus |
| | Type two diabetes mellitus |
| | Diabetic medications |
| Diagnoses and comorbidities | Hypertension |
| | Hyperthyroidism |
| | Hypothyroidism |
| | Ischaemic heart disease |
| | Heart failure |
| | Ischaemic stroke |
| | Haemorrhagic stroke |
| | All cancers |

## Statistical methods

Statistical calculations were performed using Stata 17.0. Within-individual variance was estimated using a random effects model ('mixed' command). A coefficient of variation was then calculated using the formula CV = standard deviation/mean, expressed as a decimal fraction. 95% confidence intervals were estimated from the model in Stata. Participants with identical HbA1c results on at least four different occasions were considered improbable and likely had duplicated results, so were excluded. If more than one measurement was performed on the same day, only the first result was included on the basis that others recorded for the same day were likely to be duplications or errors.

Minimal clinically important difference (MCID) was considered to be 5.5 mmol/mol (0.5%) [18]. The chance of getting a difference of at least this amount between the first and second results was calculated as a percentage. This was achieved by counting the number of participants in which every combination of first and second results was achieved. The probability of each combination of first and second results was then calculated by dividing the number of participants with that combination by the total number of participants who had the first result.

Subgroup analyses were performed by repeating the analysis on subsets of the data: by age, sex, ethnicity, Townsend neighbourhood deprivation score quintiles, (which give a score from 1 to 5 estimating deprivation in a local area, with 1 being the least deprived and 5 being the most deprived [19]), smoking and alcohol status, geographical location, year, quarter, time between measurements, mean HbA1c result, presence of prediabetes, diabetes mellitus types 1 and 2 and other comorbidities and use of diabetic medication. Further subgroup analyses were performed for participants divided according to the value of multiple variables (for example sex male, ethnicity white, <20 years age, no diabetes), with the

variables selected being those that had the most important effects on variability based on univariable analysis and which were hypothesised to be most likely clinically to have an effect.

Age and sex data were complete. Where a code for a comorbidity was absent, it was assumed not to be present. For missing ethnicity or deprivation score a missing category was present in the data. For BMI, the data appeared to be complete but in fact had a large number of implausible values – these were transformed into a missing category. The analysis of subgroups based on multiple variables included only those without missing data.

Sensitivity analyses were performed to assess bias due to factors which might affect the robustness of the results such as which statistical methods were used, number of repeat measurements included and unit of measurement. Therefore, analyses by year and quarter were performed to look for changes in population variation over time or with season. Analyses were also performed by unit (mmol/mol v percentage) to check for differences in variation due to method or mathematical factors, by number of measurements for each individual patient (calculating the CV for patients with 2, 3, 4 etc measurements each) and by inclusion or exclusion of implausible values. A simple arithmetic method of calculating CV was also performed in which the standard deviation of each individual and the mean of each individual were used to calculate an individual $CV_T$, which was then averaged using the mean. CV was also calculated using log-transformed data.

CV is expressed as a decimal fraction unless otherwise stated.

Statistical code for Stata is available from the author on request.

## Results

### Participants

11,737,653 participants registered in 832 practices were available in the database for analysis. Fig 1 shows the flowchart of participant selection. 587,023 participants had at least four measures of HbA1c recorded as mmol/mol with values between 20.1 (4%) and 195.1 (20%) mmol/mol. The median number of measurements per individual was 11 (IQR 7–18).

### Descriptive data and outcome data

Table 2 shows the participant characteristics and main univariable subgroup analyses. 587,023 participants were included in the study, with a median age at first measurement of 70 (IQR 58–79), 51.7% being male. The median number of years between first and last measurement in the database was 5 (IQR 3–7).

The overall coefficient of variation for HbA1c was 0.200 (95% CI 0.200 to 0.200), with a mean HbA1c of 52.31 mmol/mol. Variation increased with mean HbA1c levels up to moderately high mean HbA1c levels and with median HbA1c to markedly high median HbA1c levels (Fig 3 and S13 Fig in S1 File).

Table 2 and S1 Table in S1 File shows the variation of HbA1c according to diabetic status and diabetic medication. Variation was much higher in participants with prediabetes or diabetes compared to those without, and much higher in participants treated with anti-hyperglycaemic drugs than those not treated with these drugs.

Diabetic status, deprivation score, ethnicity, sex and age appear to have important effects on CV, so these were chosen for multi-subgroup analysis. The results of this are presented in a heat map in Fig 2. It can be seen from this that of these subgroups, diabetic status seems to have the most marked effect on $CV_T$.

### Subgroup analyses

Higher $CV_T$s were seen in the following subgroups: males compared to females; ex-drinkers compared to current drinkers or teetotallers; current smokers compared to ex-smokers or participants who have never smoked; middle-aged compared to older or younger participants; obese and underweight participants compared to normal weight and overweight participants; participants of black ethnicity compared to other ethnicities. Little difference in $CV_T$ was seen between patients with different comorbidities except prediabetes or diabetes.

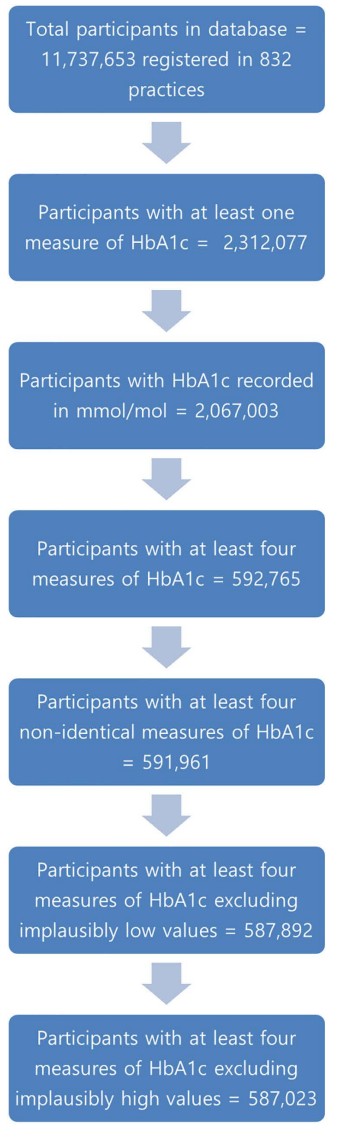

**Fig 1. Participant selection algorithm.**

Table 2 and S1 Table and S1–S9 Figs in S1 File show the major subgroup analyses.

Fig 4 shows a heat map with data on the probability of the result of a second HbA1c test given the result of an initial HbA1c test.

## Sensitivity analyses

Sensitivity analyses were performed for $CV_T$ by year, quarter, number of measurements, using different thresholds for removing participants with implausible values, median HbA1c level, log mean HbA1c level, number of days between measurements, number of days between measurements stratified by diabetes status, results coded as mmol/mol v % and a crude estimate of $CV_T$ using a simple calculation for each individual rather than a linear regression model for the group (See S2 Table and S10–S19 Figs in S1 File). There was no evidence of marked seasonality or a trend over years for the

**Table 2. Sociodemographic data and main univariable subgroup analyses.** Note that the total for diabetic status is 584,961, because some patients were excluded from this analysis because they had fewer than four results before or after a change in diabetic status. Mean HbA1c is in mmol/mol. DM=diabetes mellitus, T1DM=type 1 diabetes mellitus, T2DM=type 2 diabetes mellitus.

| | | N (%) | Mean Hba1c | $CV_T$ (95% CI) |
|---|---|---|---|---|
| All | | 587023 (100) | 52.31 | 0.200 (0.200 to 0.200) |
| Sex | Male | 303,622 (51.7) | 53.71 | 0.201 (0.201 to 0.202) |
| | Female | 283,401 (48.3) | 50.80 | 0.198 (0.198 to 0.199) |
| Age | <=10 | 259 (0.0) | 62.23 | 0.178 (0.171 to 0.184) |
| | 11–20 | 1,949 (0.3) | 63.74 | 0.209 (0.205 to 0.212) |
| | 21–30 | 7,474 (1.3) | 57.77 | 0.218 (0.125 to 0.220) |
| | 31–40 | 18,434 (3.1) | 52.66 | 0.225 (0.224 to 0.227) |
| | 41–50 | 45,032 (7.7) | 52.75 | 0.225 (0.224 to 0.226) |
| | 51–60 | 97,285 (16.6) | 53.95 | 0.221 (0.220 to 0.221) |
| | 61–70 | 133,560 (22.8) | 52.75 | 0.206 (0.205 to 0.206) |
| | 71–80 | 152,239 (25.9) | 51.11 | 0.188 (0.188 to 0.189) |
| | 81–90 | 101,672 (17.3) | 51.26 | 0.180 (0.180 to 0.181) |
| | 91–100 | 27,945 (4.8) | 51.79 | 0.179 (0.178 to 0.180) |
| | >100 | 1,174 (0.2) | 53.10 | 0.181 (0.177 to 0.185) |
| Ethnicity | Black | 10,971 (1.9) | 53.16 | 0.228 (0.226 to 0.230) |
| | Mixed | 2,303 (0.4) | 51.91 | 0.212 (0.208 to 0.216) |
| | Others | 8,221 (1.4) | 52.66 | 0.192 (0.190 to 0.194) |
| | South Asian | 25,195 (4.3) | 53.07 | 0.184 (0.183 to 0.184) |
| | White | 304,062 (51.8) | 51.97 | 0.201 (0.201 to 0.201) |
| | Missing | 236,271 (40.2) | 52.62 | 0.200 (0.200 to 0.200) |
| BMI | <18.5 | 3,531 (0.6) | 47.28 | 0.202 (0.199 to 0.205) |
| | 18.5 to <25 | 107,222 (18.3) | 48.63 | 0.181 (0.181 to 0.182) |
| | 25 to <30 | 179,744 (30.6) | 51.58 | 0.189 (0.189 to 0.189) |
| | 30 to <35 | 116,600 (19.9) | 54.48 | 0.201 (0.201 to 0.201) |
| | >35 | 80,052 (13.6) | 56.98 | 0.217 (0.216 to 0.217) |
| | Missing | 99,874 (17.0) | 51.40 | 0.207 (0.207 to 0.208) |
| Deprivation score | 1 | 103,029 (17.6) | 51.20 | 0.185 (0.185 to 0.185) |
| | 2 | 99,045 (16.9) | 52.14 | 0.191 (0.190 to 0.191) |
| | 3 | 104,133 (17.7) | 52.79 | 0.199 (0.198 to 0.199) |
| | 4 | 96,380 (16.4) | 53.83 | 0.207 (0.206 to 0.207) |
| | 5 | 72,831 (12.4) | 54.66 | 0.215 (0.215 to 0.216) |
| | Missing | 111,605 (19.0) | 50.19 | 0.204 (0.203 to 0.204) |
| Diabetic Status | No DM | 148,870 (25.4) | 38.24 | 0.091 (0.091 to 0.091) |
| | PreDM | 73,649 (12.6) | 48.74 | 0.180 (0.179 to 0.180) |
| | T1DM | 33,404 (5.7) | 70.17 | 0.157 (0.156 to 0.157) |
| | T2DM | 329,038 (56.2) | 58.70 | 0.194 (0.193 to 0.194) |

$CV_T$. An increase in $CV_T$ with number of measurements was seen up to approximately 9 repeat measurements after which the $CV_T$ plateaued. No major differences were seen using different values of cut-off points for implausible values. There was no major difference between $CV_T$ stratified by mean or median, except at extremely high or low values where median $CV_T$ was higher than mean, and no major difference when the data was log-transformed. $CV_T$ decreased with increasing median days between measurements, but when this was stratified by diabetic status, $CV_T$ decreased with increasing

| Diabetic status | Deprivation | BMI | Male | | | | Female | | | |
|---|---|---|---|---|---|---|---|---|---|---|
| | | | 20-60 | | >60 | | 20-60 | | >60 | |
| | | | White | Non-white | White | Non-white | White | Non-white | White | Non-white |
| No diabetes | 1 to 3 | <30 | 9.5 | 12.3 | 7.7 | 7.7 | 6.2 | 7.8 | 6.5 | 7.2 |
| | | >30 | 13.1 | 13.4 | 11.1 | 9.5 | 9.5 | 11.2 | 8.1 | 8.8 |
| | 4 to 5 | <30 | 10.9 | 12.9 | 8.1 | 9.7 | 9.8 | 8.5 | 8.3 | 7.9 |
| | | >30 | 17.3 | 14.2 | 15.3 | 12.3 | 11.0 | 10.6 | 9.8 | 9.6 |
| Diabetes | 1 to 3 | <30 | 17.6 | 12.8 | 11.5 | 12.1 | 15.8 | 14.0 | 9.0 | 10.7 |
| | | >30 | 19.5 | 20.6 | 14.3 | 13.9 | 16.4 | 17.9 | 12.9 | 10.1 |
| | 4 to 5 | <30 | 18.4 | 17.5 | 10.7 | 11.6 | 15.5 | 13.6 | 10.3 | 11.0 |
| | | >30 | 23.0 | 19.2 | 18.3 | 16.6 | 19.3 | 18.1 | 11.4 | 14.6 |

**Fig 2. Heat map showing the coefficient of variation expressed as % for multiple subgroup analysis according to sex, age, ethnicity, deprivation and BMI.** *Age is in years; deprivation score is Townsend deprivation score. Total N aged 20-60 = 46,275; Total N aged >60 = 83,061. Age < 20 excluded.*

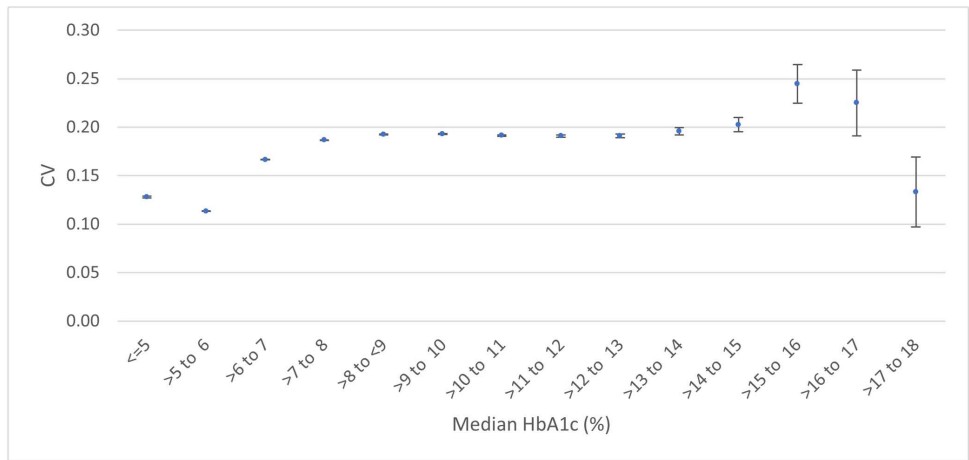

**Fig 3. Coefficient of variation by patient median HbA1c (%) with 95% confidence intervals.**

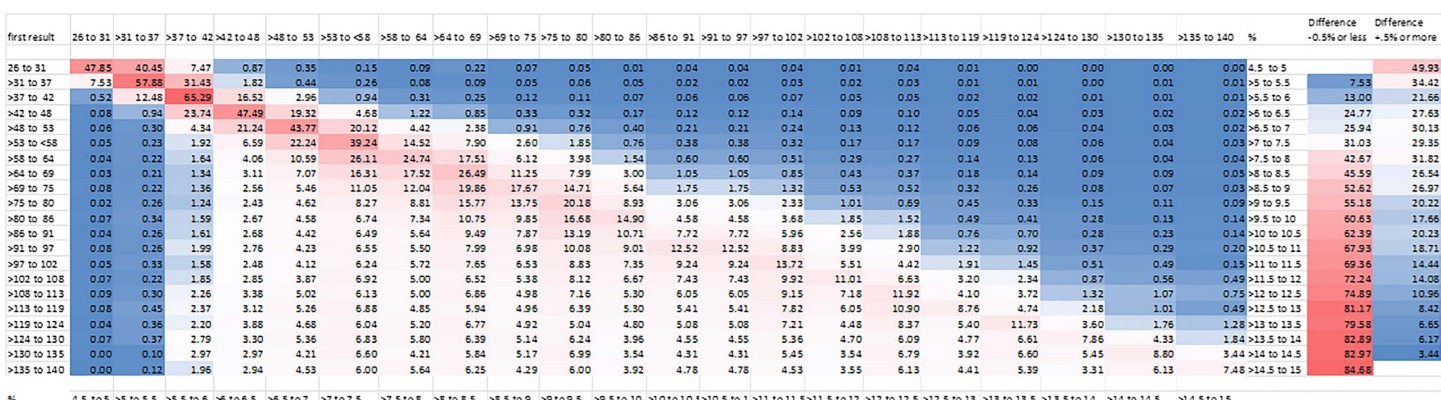

**Fig 4. Probability of second result being 5 mmol/mol or more different than first result.** *The last two columns show the probability of a second result being at least 0.5% less or more than the previous result. All age groups were included in this analysis.*

median days between measurements for non-diabetic participants but not for diabetic participants. The $CV_T$ was higher for participants with HbA1c reported as mmol/mol than as %, but if the mmol/mol were converted into % and then the $CV_T$ was calculated, the $CV_T$ between this group and the group originally reported as % was similar. This is probably because coefficient of variation is not invariant to transformations that involve adding or subtracting a constant [20]. Calculation of $CV_T$ by a crude arithmetic method resulted in a lower value than using the linear regression method.

## Discussion

### Key results

This is the largest single study of variation of HbA1c, with 587,023 participants, more than all the participants put together in the most recent systematic review of HbA1c variation [10]. The overall total within-individual coefficient of variation ($CV_T$) for this population was 0.200 (95% CI 0.200 to 0.200) which is almost three times higher than the median reported in the systematic review (0.070). This study found the $CV_T$ for non-diabetic patients was 0.091 (0.091 to 0.091), over five times higher than in the systematic review (0.017) and the EFLM biological variation database (0.016) [9]. In this study, participants with diabetes or prediabetes had much higher variation than those without these conditions. For patients with T1DM, the $CV_T$ of HbA1c in this study was 0.157 compared to 0.084 for the systematic review, and for T2DM the $CV_T$ of HbA1c was 0.194 compared to 0.083 in the systematic review – for both types of diabetes a finding of roughly twice the variation of HbA1c compared to that previously published. Note that the overall $CV_T$ includes within-individual variation in patients before and after diagnosis of DM, so it includes the long-term increase in HbA1c from non-diabetic to diabetic status. It is therefore higher than when within-individual variation is calculated for a single category of diabetic status. Total within-individual variation ($CV_T$) by diabetic status may be a more useful measure than biological variation ($CV_I$) when considering clinical implications.

CV$_T$ increased with the median HbA1 (Fig 3) up to values of around 75 mmol/mol (9%). This is important because patients with more severe disease or who are less well-controlled may have a higher variation, making management recommendations more challenging in this subgroup. Participants treated with anti-hyperglycaemic drugs had much higher variation than those not treated with these drugs. Male sex, black ethnicity, ex-drinkers, current smokers, high BMI and higher deprivation levels were associated with higher HbA1c variation. The increased variation in ex-drinkers may be caused by those who had to give up drinking for health reasons. Smoking is known to be associated with risk of diabetes onset and complications [21] Sex, BMI and deprivation may be independent risk factors for increased variability and there is evidence that ethnicity and social factors affect diabetes outcome [22] but there may be some interaction between the subgroups, for example BMI, deprivation and diabetic status being interdependent. It should also be noted that the median age of subjects was quite high at 70, suggesting this population may be more prone to pathological variation due to an increased risk of diabetes and other diseases. Multiple subgroup analysis (Fig 2) shows that diabetic status is the most important determinant of variation. Note that in this analysis, participants under 20 years of age were excluded because of the low numbers, making multiple subgroup analysis unreliable. Since type I diabetes mellitus is more common in younger patients, this might have an effect on the variability. Fig 4 shows the probability of a second measurement being different from the first measurement by 5 mmol/mol (approximately 0.5%) – the minimum clinically important difference (MCID). This shows that for every plausible initial value of HbA1c it is common for the second measurement to differ from the first by at least the MCID. Since all ages groups were included for this analysis, the implications of this table may be different for children and adolescents than adults because of the higher prevalence of type I disease in younger patients.

The CV was higher when HbA1c was reported as mmol/mol compared to reported as %, but if CV was calculated after converting mmol/mol to %, the $CV_T$ between the two % groups was similar. The difference between CV reported in % compared to mmol/mol is probably related to the fact that the conversion equation has an intercept, since the intercept does not vary. As noted in the results section, the coefficient of variation is sensitive to transformations that involve adding or subtracting a constant [20]. This will be more important at low values – at higher values the effect of the intercept on the

variation will be minimised. This problem with calculating the coefficient of variation on HbA1c reported in units of % does not appear to be widely recognised.

### Strengths and limitations

The major strengths of this study are the large size of the dataset and that the data is real-world data taken from the clinical records of primary care practices in the UK, and not measurements taken under ideal conditions. The Biological Variation Data Critical Appraisal Checklist (BIVAC) recommends biological variation studies are performed on patients in steady state, with "preanalytical procedures described and standardized to minimize preanalytical variation" [23]. This is important when using biological variation data to set analytical performance specifications, but is less relevant to clinical decision-making. The real-world nature of the data is important because these are the results on which clinical decisions are made. As these data are the results seen by clinicians, they include every component of variation (pre-analytic, analytic, biological and pathological). They therefore include variation due to changes in patient status such as diabetic status, weight and medication. This is less of an issue for those not on medication (for example non-diabetic or prediabetic) and those where measurements are closer together in time. The CV is actually higher in participants with measurements closer together in this study, but this may be an example of confounding by indication, that is participants that have less well-controlled diabetes are more likely to have more frequent measurements. Similarly, the higher variation seen in participants with higher mean HbA1c values or with diabetes could also be linked to more frequent measurement. Nevertheless, the large disparity between this study and previously published results and the large number of participants helps to compensate for this problem, and it is biologically plausible that patients with diabetes and higher mean HbA1cs have higher variation.

Assumptions were made for the repeat results calculation, namely that the change between first and second result is representative of changes in the result in subsequent calculations, which may not be the case.

Another weakness with the data is error associated with data entry and recording into the clinical databases. Data cleaning prior to analysis partially compensates for this, (as well as the fact that most lab results are transmitted electronically). For example implausible values were removed, taking the cut-off points from published data (for example [16]) and analysis of the IMRD data itself, removing of duplicate results on the same day, elimination of participants with less than four repeated measurements, and then removing any patients with four identical results which is a highly unlikely finding. Note that removing implausible values reduces the estimated variation and yet the estimated variation remains very large.

Inclusion of the entire dataset meant the only potential bias introduced into the results (that was not already present in the database) could come from data cleaning (for example exclusion of implausible values) and the specification that at least four measurements must be included. However, this latter specification, although aimed at increasing the accuracy of the variation calculation, could introduce confounding by indication since patients with more frequent tests might have more severe/less stable disease and hence a higher degree of pathological variation. Furthermore, it has been shown that the ordering of any test by a clinician increases the risk of the disease which the test is investigating, since the presence of a test in a patient record suggests there was a suspicion that patient was at risk of the disease. This could cause some degree of selection bias in the full dataset when compared to results obtained from healthy populations in controlled biological variation studies [24].

Inclusion of children and adolescents in the overall analysis might affect the results due to differences in disease type and clinical course in younger compared to older patients and S3 Fig in S1 File shows there is some variation in $CV_T$ with age.

The clinical course of the patients is not known. Most of the patients were classified as type 2 diabetes which is progressive. However, the course of this disease is variable, and some patients need intensification of treatment while others remain stable for many years These factors can cause marked HbA1c variation over long time periods.

Note that since haemoglobin glycosylation depends on the life span of red blood cells factors such as malignancy, iron deficiency, menstrual cycle, bowel diseases, some vitamin deficiencies, haemoglobinopathies, chronic kidney and/or liver disease and increasing age may influence the HbA1c values. Kidney disease and cancer as comorbidities and sex and age were considered in subgroup analyses and with the exception of age (likely associated with incidence of diabetes), made little difference to the results. Further studies could involve excluding patients with any of the above comorbidities. Analysing the mean duration of follow-up may also be helpful.

## Interpretation and generalisability

This study shows that the estimated measured total within-individual variation of HbA1c in this population of primary care patients in the UK is much higher than results for variation previously reported in the literature. This may be because the participants were less likely to be in a steady state, and the pre-analytical testing conditions were not rigorously controlled. This means that the results are likely to be more generalisable to a primary care population.

The findings of this study have clinical implications for diagnosis and monitoring. NICE guidelines recommend a diagnosis of diabetes is made if appropriate symptoms and a single measurement of HbA1c of more than 48 mmol/mol (6.5%) is present, or if asymptomatic, two measurements of HbA1c of more than 48 mmol/mol (6.5%) [11]. However, as Fig 4 shows, the chance of a patient with an initial HbA1c measurement of 48 mmol/mol to 53 mmol/mol (6.5% to 7%) having a second HbA1c level of 5.5 mmol/mol (0.5%) less is 0.26, suggesting around one in four patients with an initial HbA1c consistent with diabetes will by chance have a second measurement not consistent with this diagnosis. False positives due to not taking within-individual variability into account could lead to a patient being incorrectly diagnosed with prediabetes (which might cause stress to the patient and affect non-clinical aspects of the patient's life such as the ability to get insurance) or diabetes, which might lead to incorrect treatment. False negatives could lead to a delay in diagnosis and interventions which might increase the risk of progression and complications of the disease.

Although the management of diabetes is complex and requires multiple factors to be taken into account, in general guidelines recommend aiming for an HbA1c of less than 53 mmol/mol (7%). The $CV_T$ of this study can be used to calculate the probability with which a measured value reflects the true mean of the patient, (assuming normal distribution and using a Z-score of difference between means/standard deviation). If we take a hypothetical patient with a true HbA1c of 58 mmol/mol (7.5%), the $CV_T$ from the data in this study is approximately 0.182. This suggests a probability of 0.60 that the measured result will be 5.5 mmol/mol higher or lower than the true mean, which is the minimal clinically important difference [18]. This could be the difference between recommending only lifestyle modifications and intensifying drug treatment according to NICE guidelines [25]. Note that these guidelines apply to adults and the implications for the management of children and adolescents with diabetes are likely to be different from those for adults.

It has been previously shown that practitioners, particularly nurses, but to some extent doctors, are unaware of the inherent variation in HbA1c and adjust therapy based on very small changes in HbA1c [18]. It can thus be seen that not taking within-individual variation into account may lead to false conclusions about the magnitude and direction of change of HbA1c which could lead to appropriate interventions not being given if the measured value is higher than the real value by a clinically significant amount, leading to poor diabetes control. This would be particularly important if the real change was positive and the measured change was negative. Conversely, if the measured value was lower than the real value by a significant amount this could lead to interventions being given inappropriately with an increased risk of adverse effects, and this is particularly important if the real change was negative and the measured change was positive.

This study emphasises that clinicians should be aware of the variation of HbA1c and it demonstrates a relatively high probability of misdiagnoses or incorrect inferences, for example with regard to management of diabetes when relying on single or small numbers of measurements of HbA1c. Further, authors of clinical guidelines should take variation into account when making diagnosis, monitoring and treatment recommendations, and researchers should be aware of the variation of HbA1c when assessing the magnitude of an effect in drug trials.

Further research that addresses confounding by a change in frequency or timing of testing with increased disease severity or variation would be useful. Selecting a cohort with no change in medication may help evaluate variation in which there is no material change in the status of the participants. Studies into how variation in this population affects clinical outcomes such as mortality and whether reducing variation with improved testing and management protocols improves outcomes would be useful. Modelling the interaction between different factors affecting variation such as age, sex, BMI or diabetic status would also be helpful, but was beyond the scope of this paper.

## Conclusions

The findings of this study suggest that the estimated measured total within-individual variation of HbA1c in the UK is much higher than results for within-individual variation previously reported in the literature. Within-individual measured total variation increased with median HbA1c up to values of around 75 mmol/mol (9%).

## Supporting information

**S1 File.  Tables S1–S3 and Figs S1–S19.**
(DOCX)

## Acknowledgments

Drs Krishna Ghokale, Dr Anuradhaa Subramanian, Dr Siang Ing Lee and Dr Naijie Guan for assistance with the use of Dexter to extract data from the IMRD and CPRD databases. AS is supported by the NIHR Birmingham BRC. The views expressed are those of the author(s) and not necessarily those of the NIHR or the Department of Health and Social Care.

## Author contributions

**Conceptualization:** Alex Gough, Tom Marshall, Alice Sitch.

**Data curation:** Alex Gough.

**Formal analysis:** Alex Gough.

**Investigation:** Alex Gough.

**Methodology:** Alex Gough, Tom Marshall, Alice Sitch.

**Project administration:** Alex Gough.

**Supervision:** Tom Marshall, Alice Sitch.

**Visualization:** Alex Gough.

**Writing – original draft:** Alex Gough.

**Writing – review & editing:** Alex Gough, Tom Marshall, Alice Sitch.

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
