## [Decision Letter · Decision Letter 0]

11 Jun 2025

PONE-D-25-19490Within-individual Variation of HbA1c measurements in primary care: a retrospective cohort studyPLOS ONE

Dear Dr. Gough,

Thank you for submitting your manuscript to PLOS ONE. After careful consideration, we feel that it has merit but does not fully meet PLOS ONE’s publication criteria as it currently stands. Therefore, we invite you to submit a revised version of the manuscript that addresses the points raised during the review process.

**ACADEMIC EDITOR: ** While the manuscript exhibits intriguing potential, it requires substantial revisions and further refinement.

Although the inherent interest in the subject matter is acknowledged, the reviewers have raised crucial concerns that must be properly addressed.

We look forward to receiving your revised manuscript.

Kind regards,

Marcelo Arruda Nakazone, M.D., Ph.D.

Academic Editor

PLOS ONE

Reviewers' comments:

Reviewer's Responses to Questions

**Comments to the Author**

1. Is the manuscript technically sound, and do the data support the conclusions?

Reviewer #1: Partly

Reviewer #2: Yes

Reviewer #3: Partly

2. Has the statistical analysis been performed appropriately and rigorously? 

Reviewer #1: I Don't Know

Reviewer #2: Yes

Reviewer #3: Yes

3. Have the authors made all data underlying the findings in their manuscript fully available?

Reviewer #1: No

Reviewer #2: Yes

Reviewer #3: Yes

4. Is the manuscript presented in an intelligible fashion and written in standard English?

Reviewer #1: No

Reviewer #2: Yes

Reviewer #3: Yes

5. Review Comments to the Author

Reviewer #1: Within-individual Variation of HbA1c measurements in primary care: a

retrospective cohort study

Congratulate to the authors for undertaking a mammoth task reporting on real-world data collected over a decade. There is no doubt that findings from this study will impact clinical management and interpretation of test results. The reviewer offers comments with the aim to support the publication of a sound and impactful manuscript.

General comments

Writing / spelling / abbreviations

It seems “participants” is missing after 11,737,653 in “… dataset used for this study consisted of 11,737,653 registered ….”

Authors are encouraged to thoroughly revise their manuscript to ensure coherent sentences [i.e “…more than the all the participants put …” (sentence in Results)].

Revise “i.e” in “… patient mean HbA1c i.e. with more …” (in Abstract) and elsewhere in the manuscript. Authors also use “e.g.” later in the manuscript. Consider rather writing the word(s), like authors have also done in the manuscript, for consistency and clarity.

Ensure correct and appropriate use of capital letters throughout the manuscript, including references to Tables, Figures and corresponding titles / legends.

Please be consistent in spelling, for instance, “… more consistent with units of

per cent, suggesting most of these low values were actually percent values erroneously recorded …” where both “per cent” and “percent” are used [unless these words have different meanings – in which case it would probably be advisable to use different words to denote the different meanings.

Please remove duplicate words in sentences, for instance, “This was achieved by counting the number the number of participants in which every combination of first and second results was achieved” (in Statistical methods) and “The findings of this study suggest that the estimated measured total within-individual variation of HbA1c in in the UK is much higher …” (in Conclusion) including throughout the manuscript .

Abstract

Authors do not mention the research study design in the abstract.

Please confirm that the zero-width CI reported here “(95%CI 0.20 to 0.20)” [or as stated in Results - (95%CI 0.200 to 0.200)] is correct, especially given the statement “Note that the overall CVT includes within-individual variation in patients before and after diagnosis of DM, so it includes the longterm increase in HbA1c from non-diabetic to diabetic status” which creates an expectation of large(r) CI width? This question applies to most, if not all CIs reported, even those with slight sizes.

Are there words missing in this sentence: “Weaknesses included difficulties accounting for confounding by indication”. If no words are missing, please rephrase to clarify what is said.

Aim

It is unclear what authors mean with “…resulting in clinically important changes in HbA1c” in the sentence “To describe the probability of within-individual variation in HbA1c resulting in clinically important changes in HbA1c”. Does this mean the aim was to determine within-individual variation [WIV] to inform clinical decision-making / utility related to observed HbA1c changes in a patient. Or do authors mean describing probability of WIV will inform clinically important changes in HbA1c [meaning the test/test result] itself?

Methods

Setting

Irrespective of data obtained “from database inception” until the date of last entry (11th January 2021) ultimately, authors report on data analysed for the period 2010-2019, thus data spanning a decade. Authors are encouraged to provide the reader with this timeframe as this perspective underscores “This is the largest single study of variation of HbA1c, with 587,023 participants, more than the all the participants put together in the most recent systematic…” and “The major strengths of this study are the large size of the dataset and that the data is realworld data taken from the clinical records of primary care practices …” [revise realworld to real-world]

Authors need to clarify why 2020 data are not included in the analysis, given last data entry date was 11th January 2021.

Statistical Methods

Should “age young” be specified given that 10-year age increments are reported, which raise the question whether <20 years equals “age young”.

Results

Age in Figure 2 starts at 20 years [see additional comment under Table 2]. A footnote to address age groups included / excluded in this analysis would be helpful, including addressing possible result differences, or not, if all age groups were included in this analysis. Alternatively, would it be more informative to offer a similar Figure 2 analysis for <20 years?

Figure 4: labelling of rows and columns are indecipherable while most numbers are indistinguishable. The reviewer assumes that this will be taken care of when manuscript is published. A sentence such as “However, as figure 4 shows, the chance of a patient with an initial HbA1c measurement of 48mmol/mol to 53mmol/mol (6.5% to 7%) having a second HbA1c level of 5.5mmol/mol (0.5%) less is 0.26, suggesting around …” thus becomes impossible to verify against Table 4. Please specify whether all age groups were included in Table 4 analysis? If all age groups were included, would it be necessary to address clinical implications with regards to children and adolescence in this paragraph. Authors reference two adult specific NICE guideline documents in the manuscript. It is thus assumed that any text relates to clinical implications for diagnosis and monitoring of adults (only). Would it be necessary to distinguish whether all statements apply to children and adolescents as well. Would it be necessary to reference NICE guidelines for children and adolescents. Alternatively, would it be more informative to address specifics with regards to within-individual Hb1Ac variation regarding children and adolescents.

Strengths and Limitations

Would authors consider inclusion of children and adolescence [assumed as the <20 years group] as a possible limitation should one accepts that such cohort-specific analysis might have yielded different within-individual variation results. However, this might have been covered in sub-group analyses, yet not explicitly stated – statements which will be enhance the manuscript. This comment is furthermore linked to what appears as a disproportionate low number of children and adolescence [the <20 years group] included in the study, covering a decade (gauged against reported UK incidence / prevalence for children and adolescence). This might need to be addressed.

Table 2:

Age groups: as currently delineated it is assumed age groups imply 0-10 years, 11-20 years, 21-30 years, 31-40 years etc. Yet, in Figure 2, analysis includes age 20 years. Does this imply, for Figure 2 analysis, age groups as per Table 2 were “ignored” to include x number of 20year old participants in the analysis. If this is the case authors would need to add a footnote to Figure 2 to indicate the number of participants included in the category 20-60 years as a reader would not be able to calculate this number for themselves, using Table 2. Also, it might be more accurate to offer actual ages per category [for instance 21-30 instead of >20-30] to leave no room for confusion / interpretation. For instance, in age groups >80 to 90 and 90 to 100, to which group were participants aged 90 years allocated. With regards to the 90 to 100 group – does this mean no participants were 101, 102, 103 etc years.

Age: total count = 585849. Advise as to reason for not reporting on 587023 participants. Revise percentage calculations or add footnote to clarify.

Diabetic status: total count = 584961 yet denominator used to calculate percentage = 587023. Advise as to reason for not reporting on 587023 participants. Revise percentage calculations or add footnote to clarify.

Supplementary Materials

Table S1:

Co-morbidities – authors used 587023 as denominator to calculate percentages except for PDM, DMI1 or DM2 …; heart failure; heart disease. Either revise or add footnote to clarify, including denominator used for particular co-morbidities.

Diabetic medication: total count = 461914 yet denominator used to calculate percentage = 587023. However, not all percentages shown correspond to a denominator of 587023 though. Advise as to reason for not reporting on 587023 participants. Revise percentage calculations or add footnote to clarify, including denominator used if different from overall number of participants.

Region: total count = 441808 yet denominator used to calculate percentage = 587023, except for South East Coast and Wales [denominator used unclear]. Advise as to reason for not reporting on 587023 participants. Revise percentage calculations or add footnote to clarify, including denominator used in different from overall number of participants.

Alcohol consumption: total count = 475285 yet denominator used to calculate percentage = 587023. Advise as to reason for not reporting on 587023 participants. Revise percentage calculations or add footnote to clarify.

Smoker status: total count = 523943 yet denominator used to calculate percentage = 587023. Advise as to reason for not reporting on 587023 participants. Revise percentage calculations or add footnote to clarify.

Authors need to confirm that every statistic provided in Tables 2 and S1, and throughout the manuscript and supplementary materials were verified and are correct.

Figure S11: Please revise “Chart Title”

Authors might want to consider adding text specifying applicable unit(s) of measurement(s) across Figures and/or specific to Figures. Alternatively, add the unit(s) of measurement to each figure.

Reviewer #2: PONE-D-25-19490

Registry based study focuses on individual HbA1c variability during a course of time. Data is UK based and comes from general practitioners. Total number of patients was over 11 million and just over 0.5 million was included for this analysis. Variability of HbA1c was three to five times higher than previously known (as quoted by previous extensive literature review). Variability over time was as high as 20% and it was higher if type 2 diabetes. Patients with diabetes of any degree showed widest variability. Database analysis leaves questions. Some clarifications need at least discussion.

When was the used database initially set up? End date for the analysis is given but not the start. Five years before cut-off point of 2021 for all patients?

What analysis methods (HbA1c) were used? HPLC or immunologic? Was there a change from one HbA1c method to another at some point?

Was POC analysis used or only central laboratory? Quality control? Was there some kind of centrally organized evaluation of laboratory process?

Was unit change (percentage to mmol/mol) done during the study follow-up? If so, at the same time with all centers/database?

Database included 150000 patients without pre- or clinical diabetes. What is the background for repeated HbA1c measurements for considerable time span? Screening from some clinical reason?

Clinical course of the patients is not known. Most of the patients were classified Type 2. Clinical course of type 2 diabetes is progressive. While pathogenesis is mostly unknown, several clinically different courses do exist. Even subtyping is possible (Ahlqvist et al. Lancet Diabetes Endocrinol. 2019;7:668-669. doi: 10.1016/S2213-8587(19)30257-8). Some patients need intensification of the treatment while some others may remain stable for years. Both factors cause considerable variation of HbA1c over time.

Degree of hemoglobin glycosylation depends on the life span of red blood cells. Please discuss factors like iron deficiency, menstrual cycle, bowel diseases, some vitamin deficiencies, malignancies etc. These factors and advancing age may influence HbA1c values.

Minor: Some minor typos (like page 13 third line …the number the number… Discussion first line)

Figure 4 legend …5 mol/mol…

Table 2 HbA1c is given in % while most text uses mmol/mol.

Otherwise, text is clear. There are some repetitions. Database hygiene, avoiding biases or mistakes, selecting factors for analysis and statistics is OK.

Reviewer #3: This study explores variation in HbA1c levels in a large cohort. Few comments:

1) Apart from the larger sample size and the stronger results, it is unclear what this study brings to the literature. A higher variation of HbA1c is expected at higher levels, as individuals are likely to receive treatment and/or intensive monitoring to support their diabetes care. Indeed, diabetes diagnosis was the strongest predictor of variation.

2) The time between measurements and mean duration of follow-up is not mentioned.

3) Conditions and/or diseases that may be directly associated with inaccuracies in HbA1c measurements, such haemoglobinopathies, spleen disorders, anaemia, chronic kidney and/or liver disease, are not included as confounders. Analyses should controlled for such parameters.

4) Please ensure that all abbreviations and units are provided in the tables/figures. The figure legends should contain all the necessary info without the need to refer to the main text.

6. PLOS authors have the option to publish the peer review history of their article (what does this mean? ). If published, this will include your full peer review and any attached files.

**Do you want your identity to be public for this peer review?** For information about this choice, including consent withdrawal, please see our Privacy Policy .

Reviewer #1: No

Reviewer #2: **Yes: ** Jorma T Lahtela

Reviewer #3: No

---

## [Author Response · Author response to Decision Letter 1]

8 Jul 2025

2. Journal. When submitting your revision, we need you to address these additional requirements. Please ensure that your manuscript meets PLOS ONE's style requirements, including those for file naming.

Author: The headings and file names have been revised.

3. Journal: Thank you for uploading your study's underlying data set. Unfortunately, the repository you have noted in your Data Availability statement does not qualify as an acceptable data repository according to PLOS's standards.

Author: The authors do not have permission to share the data. The data availability statement has been updated.

4. Journal: Please include captions for your Supporting Information files at the end of your manuscript, and update any in-text citations to match accordingly. Please see our Supporting Information guidelines for more information: http://journals.plos.org/plosone/s/supporting-information.

Author: This has been done

5. Reviewer: Writing / spelling / abbreviations

It seems “participants” is missing after 11,737,653 in “… dataset used for this study consisted of 11,737,653 registered ….”

Authors are encouraged to thoroughly revise their manuscript to ensure coherent sentences [i.e “…more than the all the participants put …” (sentence in Results)].

Revise “i.e” in “… patient mean HbA1c i.e. with more …” (in Abstract) and elsewhere in the manuscript. Authors also use “e.g.” later in the manuscript. Consider rather writing the word(s), like authors have also done in the manuscript, for consistency and clarity.

Ensure correct and appropriate use of capital letters throughout the manuscript, including references to Tables, Figures and corresponding titles / legends.

Please be consistent in spelling, for instance, “… more consistent with units of per cent, suggesting most of these low values were actually percent values erroneously recorded …” where both “per cent” and “percent” are used [unless these words have different meanings – in which case it would probably be advisable to use different words to denote the different meanings.

Please remove duplicate words in sentences, for instance, “This was achieved by counting the number the number of participants in which every combination of first and second results was achieved” (in Statistical methods) and “The findings of this study suggest that the estimated measured total within-individual variation of HbA1c in in the UK is much higher …” (in Conclusion) including throughout the manuscript .

Author: The specific typos have been addressed and the whole manuscript has been thoroughly proofread.

6.

Reviewer: Authors do not mention the research study design in the abstract.

Author: Corrected in abstract.

7. Reviewer: Please confirm that the zero-width CI reported here “(95%CI 0.20 to 0.20)” [or as stated in Results - (95%CI 0.200 to 0.200)] is correct, especially given the statement “Note that the overall CVT includes within-individual variation in patients before and after diagnosis of DM, so it includes the long-term increase in HbA1c from non-diabetic to diabetic status” which creates an expectation of large(r) CI width? This question applies to most, if not all CIs reported, even those with slight sizes.

Author: The zero-width CI is correct due to the large sample size.

8. Reviewer: Are there words missing in this sentence: “Weaknesses included difficulties accounting for confounding by indication”. If no words are missing, please rephrase to clarify what is said.

Author: Text has been rephrased.

9. Reviewer: It is unclear what authors mean with “…resulting in clinically important changes in HbA1c” in the sentence “To describe the probability of within-individual variation in HbA1c resulting in clinically important changes in HbA1c”. Does this mean the aim was to determine within-individual variation [WIV] to inform clinical decision-making / utility related to observed HbA1c changes in a patient. Or do authors mean describing probability of WIV will inform clinically important changes in HbA1c [meaning the test/test result] itself?

Author: Additional explanation given in Aim

10. and 11. Reviewer:

Irrespective of data obtained “from database inception” until the date of last entry (11th January 2021) ultimately, authors report on data analysed for the period 2010-2019, thus data spanning a decade. Authors are encouraged to provide the reader with this timeframe as this perspective underscores “This is the largest single study of variation of HbA1c, with 587,023 participants, more than the all the participants put together in the most recent systematic…” and “The major strengths of this study are the large size of the dataset and that the data is realworld data taken from the clinical records of primary care practices …” [revise realworld to real-world]

Authors need to clarify why 2020 data are not included in the analysis, given last data entry date was 11th January 2021.

Author: This has been clarified in the text under setting – the whole database was used for analyses except the sensitivity analyses by year for the years prior to 2010 and after 2019 as these years were not complete in the database.

13. Reviewer: Should “age young” be specified given that 10-year age increments are reported, which raise the question whether <20 years equals “age young”.

Author: Corrected in statistical methods.

14. Reviewer: Results

Age in Figure 2 starts at 20 years [see additional comment under Table 2]. A footnote to address age groups included / excluded in this analysis would be helpful, including addressing possible result differences, or not, if all age groups were included in this analysis. Alternatively, would it be more informative to offer a similar Figure 2 analysis for <20 years?

Author: Addressed in fig 2 legend and Key Results.

15a. Reviewer Figure 4: labelling of rows and columns are indecipherable while most numbers are indistinguishable. The reviewer assumes that this will be taken care of when manuscript is published. A sentence such as “However, as figure 4 shows, the chance of a patient with an initial HbA1c measurement of 48mmol/mol to 53mmol/mol (6.5% to 7%) having a second HbA1c level of 5.5mmol/mol (0.5%) less is 0.26, suggesting around …” thus becomes impossible to verify against Table 4.

Author: Fig 4 has been edited in PACE and resubmitted.

15b Reviewer:Please specify whether all age groups were included in Table 4 analysis? If all age groups were included, would it be necessary to address clinical implications with regards to children and adolescence in this paragraph.

Author: Addressed in fig 4 legend and Key Results

16. Reviewer: Authors reference two adult specific NICE guideline documents in the manuscript. It is thus assumed that any text relates to clinical implications for diagnosis and monitoring of adults (only). Would it be necessary to distinguish whether all statements apply to children and adolescents as well. Would it be necessary to reference NICE guidelines for children and adolescents. Alternatively, would it be more informative to address specifics with regards to within-individual Hb1Ac variation regarding children and adolescents.

Author: Line added in Interpretation and Generalisability

17. Reviewer: Strengths and Limitations

Would authors consider inclusion of children and adolescence [assumed as the <20 years group] as a possible limitation should one accepts that such cohort-specific analysis might have yielded different within-individual variation results. However, this might have been covered in sub-group analyses, yet not explicitly stated – statements which will be enhance the manuscript. This comment is furthermore linked to what appears as a disproportionate low number of children and adolescence [the <20 years group] included in the study, covering a decade (gauged against reported UK incidence / prevalence for children and adolescence). This might need to be addressed.

Author: This has been mentioned as limitation in the discussion, and figure S3 is mentioned which shows the effect of age on CVT.

18a. Reviewer

Age groups: as currently delineated it is assumed age groups imply 0-10 years, 11-20 years, 21-30 years, 31-40 years etc. Yet, in Figure 2, analysis includes age 20 years. Does this imply, for Figure 2 analysis, age groups as per Table 2 were “ignored” to include x number of 20year old participants in the analysis. If this is the case authors would need to add a footnote to Figure 2 to indicate the number of participants included in the category 20-60 years as a reader would not be able to calculate this number for themselves, using Table 2.

Author: The reviewer is correct. Number of participants has been added to fig 2 legend.

18b. Reviewer. Also, it might be more accurate to offer actual ages per category [for instance 21-30 instead of >20-30] to leave no room for confusion / interpretation. For instance, in age groups >80 to 90 and 90 to 100, to which group were participants aged 90 years allocated. With regards to the 90 to 100 group – does this mean no participants were 101, 102, 103 etc years.

Author: Revised in Table 2

19. Reviewer: Age: total count = 585849. Advise as to reason for not reporting on 587023 participants. Revise percentage calculations or add footnote to clarify.

Diabetic status: total count = 584961 yet denominator used to calculate percentage = 587023. Advise as to reason for not reporting on 587023 participants. Revise percentage calculations or add footnote to clarify.

Author: Age has been updated to include >100 years to make the correct denominator. Clarification in heading of table 2 as to why the denominator for diabetic status is different, and percentages have been revised.

20. Reviewer: Supplementary Materials. Table S1:

Author: Thanks for spotting the errors in the percentage calculations in table S1 which were due to missing data not recorded properly in the dataset for the smoking and alcohol status calculations. The calculations have been reviewed and are now correct. A minor error in coding has also been noted and recalculations performed for table S2 ( days between measurements) which has made no material difference to the results . See further comments below:

Reviewer Co-morbidities – authors used 587023 as denominator to calculate percentages except for PDM, DMI1 or DM2 …; heart failure; heart disease. Either revise or add footnote to clarify, including denominator used for particular co-morbidities.

Author – I have checked these figures, and the calculations in the original manuscript appear correct, with 587023 used as a denominator for all comorbidities

Reviewer: Diabetic medication: total count = 461914 yet denominator used to calculate percentage = 587023. However, not all percentages shown correspond to a denominator of 587023 though. Advise as to reason for not reporting on 587023 participants. Revise percentage calculations or add footnote to clarify, including denominator used if different from overall number of participants.

Author Footnote added to clarify – the total is less due to a chance in diabetic status meaning some are excluded due to fewer than 4 results after a change in status.

Reviewer. Region: total count = 441808 yet denominator used to calculate percentage = 587023, except for South East Coast and Wales [denominator used unclear]. Advise as to reason for not reporting on 587023 participants. Revise percentage calculations or add footnote to clarify, including denominator used in different from overall number of participants.

Author – I have checked these figures, and the calculations in the original manuscript appear correct, with 587023 used as a denominator for all regions, and a total region count of 587023

Reviewer: Alcohol consumption: total count = 475285 yet denominator used to calculate percentage = 587023. Advise as to reason for not reporting on 587023 participants. Revise percentage calculations or add footnote to clarify.

Author: As noted above this was due to missing data which has now been corrected

Reviewer: Smoker status: total count = 523943 yet denominator used to calculate percentage = 587023. Advise as to reason for not reporting on 587023 participants. Revise percentage calculations or add footnote to clarify.

Author: As noted above this was due to missing data which has now been corrected

21a. Reviewer Figure S11: Please revise “Chart Title”

Author: Done

21b: Reviewer: Authors might want to consider adding text specifying applicable unit(s) of measurement(s) across Figures and/or specific to Figures. Alternatively, add the unit(s) of measurement to each figure.

Author: Legends updated and a note added at end of statistical methods to confirm CV is expressed as a decimal fraction unless otherwise stated.

22. Reviewer: When was the used database initially set up? End date for the analysis is given but not the start. Five years before cut-off point of 2021 for all patients?

Author: Answer given in study design and data sources.

23. Reviewer. What analysis methods (HbA1c) were used? HPLC or immunologic? Was there a change from one HbA1c method to another at some point?

Author: Answer given in study design and data sources.

24. Reviewer: Was POC analysis used or only central laboratory? Quality control? Was there some kind of centrally organized evaluation of laboratory process?

Author: Answer given in study design and data sources.

25. Reviewer: Was unit change (percentage to mmol/mol) done during the study follow-up? If so, at the same time with all centers/database?

Author: Answer given in study design and data sources.

26. Reviewer: Database included 150000 patients without pre- or clinical diabetes. What is the background for repeated HbA1c measurements for considerable time span? Screening from some clinical reason?

Author: This information is not available in the database

27. Reviewer. Clinical course of the patients is not known. Most of the patients were classified Type 2. Clinical course of type 2 diabetes is progressive. While pathogenesis is mostly unknown, several clinically different courses do exist. Even subtyping is possible (Ahlqvist et al. Lancet Diabetes Endocrinol. 2019;7:668-669. doi: 10.1016/S2213-8587(19)30257-8). Some patients need intensification of the treatment while some others may remain stable for years. Both factors cause considerable variation of HbA1c over time.

Author: This point has been added as a limitation in the discussion.

28. Reviewer: Degree of hemoglobin glycosylation depends on the life span of red blood cells. Please discuss factors like iron deficiency, menstrual cycle, bowel diseases, some vitamin deficiencies, malignancies etc. These factors and advancing age may influence HbA1c values.

And

32 Conditions and/or diseases that may be directly associated with inaccuracies in HbA1c measurements, such haemoglobinopathies, spleen disorders, anaemia, chronic kidney and/or liver disease, are not included as confounders. Analyses should controlled for such parameters.

Author: These conditions have been mentioned in the discussion as a limitation.

29 Reviewer: Minor: Some minor typos (like page 13 third line …the number the number… Discussion first line)

Figure 4 legend …5 mol/mol…

Table 2 HbA1c is given in % while most text uses mmol/mol.

Author: Typos corrected

30 Reviewer: Apart from the larger sample size and the stronger results, it is unclear what this study brings to the literature. A higher variation of HbA1c is expected at higher levels, as individuals are likely to receive treatment and/or intensive monitoring to support their diabetes care. Indeed, diabetes diagnosis was the strongest predictor of variation.

Author: This is the first study of its kind, we consider the larger sample size and stronger results alone contribute to the literature.

31. Reviewer. The time between measurements and mean duration of follow-u

---

## [Decision Letter · Decision Letter 1]

15 Sep 2025

Within-individual variation of HbA1c measurements in primary care: a retrospective cohort study

PONE-D-25-19490R1

Dear Dr. Gough,

We’re pleased to inform you that your manuscript has been judged scientifically suitable for publication and will be formally accepted for publication once it meets all outstanding technical requirements.

Kind regards,

Marcelo Arruda Nakazone, M.D., Ph.D.

Academic Editor

PLOS ONE

Additional Editor Comments (optional):

Reviewers' comments:

Reviewer's Responses to Questions

**Comments to the Author**

1. If the authors have adequately addressed your comments raised in a previous round of review and you feel that this manuscript is now acceptable for publication, you may indicate that here to bypass the “Comments to the Author” section, enter your conflict of interest statement in the “Confidential to Editor” section, and submit your "Accept" recommendation.

Reviewer #1: All comments have been addressed

Reviewer #2: All comments have been addressed

Reviewer #3: All comments have been addressed

2. Is the manuscript technically sound, and do the data support the conclusions?

Reviewer #1: Yes

Reviewer #2: Yes

Reviewer #3: Yes

3. Has the statistical analysis been performed appropriately and rigorously? 

Reviewer #1: Yes

Reviewer #2: Yes

Reviewer #3: Yes

4. Have the authors made all data underlying the findings in their manuscript fully available?

Reviewer #1: No

Reviewer #2: Yes

Reviewer #3: Yes

5. Is the manuscript presented in an intelligible fashion and written in standard English?

Reviewer #1: Yes

Reviewer #2: Yes

Reviewer #3: Yes

6. Review Comments to the Author

Reviewer #1: (No Response)

Reviewer #2: Clarifications and corrections to my questions acceptable.

Only minor concern is HbA1c method (Methods section). It is written “…UK laboratories moved to the HPLC method after 2010.” Most countries moved to immunologic methods due to high costs of HPLC. I was not able to check if HPLC – a golden standard – was major method in UK throughout the study period.

Reviewer #3: All comments have been addressed. A stronger conclusion highlighting the clinical implications of the study would add to the manuscript.

7. PLOS authors have the option to publish the peer review history of their article (what does this mean? ). If published, this will include your full peer review and any attached files.

**Do you want your identity to be public for this peer review?** For information about this choice, including consent withdrawal, please see our Privacy Policy .

Reviewer #1: No

Reviewer #2: **Yes: ** Jorma T Lahtela

Reviewer #3: No

---

## [Editor Report · Acceptance letter]

PONE-D-25-19490R1

PLOS ONE

Dear Dr. Gough,

I'm pleased to inform you that your manuscript has been deemed suitable for publication in PLOS ONE. Congratulations! Your manuscript is now being handed over to our production team.

Kind regards,

on behalf of

Professor Marcelo Arruda Nakazone

Academic Editor

PLOS ONE